# Mathematical Models and Data Analysis of Residential Land Leasing Behavior of District Governments of Beijing in China

Jing Cheng 

Architecture and Civil Engineering Research Centre, City University of Hong Kong Shenzhen Research Institute, Shenzhen 518057, China; jingcheng7-c@my.cityu.edu.hk; Tel.: +86-138-1825-8328

**Abstract:** To analyze the leasing behavior of residential land in Beijing, the mathematical models of the price and the total area of the leased residential land are presented. The variables of the mathematical models are proposed by analyzing the factors influencing the district government's leasing behavior for residential land based on the leasing right for residential land in Beijing, China. The regression formulae of the mathematical models are obtained with the ordinary least squares method. By introducing the data of the districts in Beijing from 2004 to 2015 into the mathematical models, the numerical results of the coefficients in the mathematical models are obtained by solving the equations of the regression formulae. After discussing the numerical results of the influencing factors, the district government behavior for leasing residential land in Beijing, China, is investigated. The numerical results show the factors concerning the government and how these factors influence the leased price and the total leased area of residential land for this large city in China. Finally, policy implications for the district government regarding residential land leasing in Beijing are proposed.

**Keywords:** mathematical model; leased price; total leased area; data analysis; residential land; Beijing



## 1. Introduction

For a country, the leasing of residential land is related to people's life and the economic development of the country. With the developments in China in the past 30 years, the land and real estate markets also have been developing rapidly, and the mode of residential land leasing has been fixed. To analyze how the mode is fixed, considering that land leasing is a kind of government behavior, it is necessary to analyze the factors influencing government behavior for leasing residential land in China to enhance the efficient use of residential land and achieve sustainable development. Therefore, studying the reasonable allocation of land resources is necessary for regional urban planning [1], land policies and planning [2], and the development of cities [3].

In China, for the metropolitans, such as Beijing, Shenzhen and Shanghai, the district governments of these cities decide the leasing behavior of the land use rights, and then land markets are mainly controlled by the district governments [4,5]. The central and local governments are responsible for managing the different parts of the urban housing and land markets [6]. Various policies are formulated by the central government to promote the marketization of land, while the local governments implement these polices to maximize long-term profits and ensure long-term sustainable development [7]. The policies for land leasing are affected by the fiscal incentives; thus, the local governments considered attracting the land investments [8]. Li et al. [9] considered multi-objective functions to discuss the optimization of land-use arrangement. Therefore, when the local governments lease land, the factors influencing land leasing should be considered [10].

Some factors affecting the land leasing of the local governments are discussed by using mathematical models. Land price [11,12], land leasing mode [13], political competition among local officials [14], the economy of the city and so on are considered. The listing, tender and auction of land leasing modes stipulated by the Chinese land law affect the

local governments' decisions on leasing land [13]. The increase in gross domestic product (GDP) of a city can make the local government increase the land leasing [12]. The behavior of land leasing of the local government and the prices of the land and houses can also be influenced by local public goods, such as public transit [15], airports [16], subway stations [17] and so on. In suburban districts of large Chinese cities, subway stations make the local governments obtain more profits from leasing land near these stations [18], because they impact the consumer amenities and economic activities positively [17]. The shorter the distances to subway stations are, the higher the land prices are [19]. Land or houses near pure air, high-quality primary and middle schools, main universities and environmental conveniences raise their prices [15]. Land prices can also be influenced by the distance between the land location and the center of a city. The shorter this distance is, the higher the prices of the land leases are [16].

For the influencing factors of leased land area, the shorter the distance of the land location to the city center is, the larger the total area of the leased land is [20]. The distance of the land to the city center or the closest subway station has a negative correlation with the total area of leased land [21]. The district location and the highway number in this district have influences on the total leased area of industrial land [5]. Furthermore, the total area of leased land is also influenced by the distance of the land to the city or district center, metro station and highway number [22].

In general, in the mathematical models of the leased land, the regression formulae are obtained with the ordinary least squares (OLS) method [23–25]. With the further development of the OLS method, the general least squares method, moving least squares method [26,27], improved moving least squares method [28,29] and complex variable moving least squares method [30] are presented. The general least squares method can also be used in the mathematical models of the leased land to obtain the regression formulae. The moving least squares method, improved moving least squares method and complex variable moving least squares method are presented to propose new meshless methods to solve science and engineering problems governed by the partial differential equations.

Beijing, as the capital, is one of the largest Chinese cities. It has a high GDP, land price as well as house price. The population of Beijing reached more than 20 million. With the rapid economic development of Beijing, the pressures on land supply increased, and the contradiction between land resources and population growth became sharp. As mentioned above, the district governments in Beijing decide the planning and leasing of residential land, which shows that in Beijing, the data of the districts in Beijing should be used to study the leasing behavior regarding residential land. Currently, the corresponding research on the influencing factors of the leasing prices for house and land are mainly based on the data of a city or province. Few papers have been published studying the factors affecting the leasing behavior of residential land of district governments in Beijing by using the data of the districts, because it is difficult to obtain the corresponding data at the district level. The corresponding research mainly considered the factors of basic land and location, while the district factors, such as political and economic influences, were not discussed; thus, more factors from different aspects should be considered. Furthermore, most of the literature discussed land leasing prices, while few researchers applied the price and the total area of leased land together to study the land leasing behavior of the local governments.

In this paper, the mathematical models of the price and the total area of the leased residential land are presented to analyze the leasing behavior of residential land in Beijing. The variables of the mathematical models are proposed based on the leasing right for residential land in Beijing, China. The regression formulae of the mathematical models are obtained with the OLS method. The data of the districts in Beijing from 2004 to 2015 are used to obtain the numerical results of the coefficients in the models. The district government's behavior towards leasing residential land in Beijing, China, is investigated by discussing the numerical results of the models.

## 2. Methodology

The methodology in this paper is to discuss the factors influencing the leasing behavior of the district governments for leasing residential land by using the mathematical models of the price and the total area of leased residential land. Linear regression relationships are applied in the models proposed in this paper, and the regression formulae of the coefficients in the models are obtained from the OLS method.

By using the data of all districts in Beijing from 2004 to 2015, numerical results can be obtained from the regression formulae of the mathematical models. From the results, the relationship between the price (or total area) of residential land leased and the corresponding influencing factors are analyzed.

The mathematical models of the price and the total area of leased residential land presented in this paper are for every piece of land for all districts in Beijing for a year. The dependent variables are residential land price and total area, and the corresponding independent ones are the factors affecting the local district governments' behavior on leasing residential land.

From the literature published before, all possible factors are considered in the mathematical models in this paper. From the macro level of a district and micro level of a piece of leased land, the land attributes, district-level economic status and geographic location of the leased land are discussed.

The regression formulae of the mathematical models are obtained from the OLS method in which the best approximation is used to obtain the most optimal solutions, and then the correct numerical solutions can be obtained from the models.

According to the data of the price and the total area of leased residential land and the corresponding influencing factors in the districts of Beijing from 2004 to 2015, the numerical results can be obtained from the regression formulae. The variance inflation factor (VIF) is applied to avoid multicollinearity among the independent variables of the mathematical models.

From the numerical results, the factors influencing the local district governments on leasing residential land are discussed, and how the price and the total area of leased residential land being changed with the factors is analyzed. Finally, policy implications for the local district governments are proposed, and the metropolitans of developing countries implementing public ownership of land can learn from and benefit from these implications.

## 3. Mathematical Models

### 3.1. The Dependent and Independent Variables

The dependent variables considered in this paper are the price and the total area of residential land leased. Then the mathematical models of the price and the total area are proposed, respectively.

The independent variables considered are the factors influencing the local district governments on leasing residential land. These factors are proposed based on the literature from the macro level of a district and micro level of a piece of leased land, such as land attributes, district-level economy, geographic location of the leased land and so on. Thus, land attributes, district-level attributes and location attributes, of factors affecting residential land leasing, are considered.

With regard to land attributes, the factors considered in this paper are the area and the floor area ratio (FAR) of the residential land, the mode of land leasing and the corresponding district location. The land area is the built area of a certain leased residential land. The FAR of the land is the ratio of the built area to the area of a piece of residential land. These two factors will be considered by the government because the area and FAR of the leased land are needed for different land leasing types. The land leasing mode can affect land leasing, and the price of land leased by listing mode can be higher [13]. The area and price of residential land are different in the various districts in Beijing. In Beijing, three categories, i.e., center districts, suburban ones and county, are considered for the 18 districts from 2004 to 2015. Districts such as Dongcheng, Xicheng, Chaoyang, Haidian, Chongwen, Xuanwu,

Fengtai and Shijingshan, are the center ones; the districts of Daxing, Changping, Fangshan, Shunyi, Mentougou, Tongzhou, Pinggu and Huairou are suburban ones, whereas Yanqing and Miyun are counties. The district location can influence land prices in that the land prices in the central districts are higher than those in other districts [16]. The geographic distribution of the 18 districts in Beijing is shown in Figure 1.

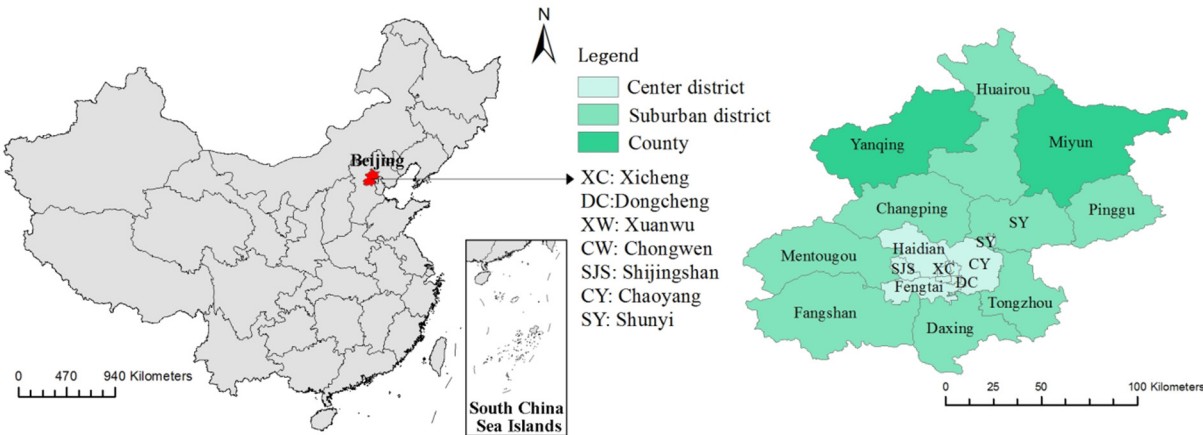

**Figure 1.** The distribution of the center districts, suburban districts and counties [3].

With regard to district-level attributes, factors such as GDP and term of office for the district head are involved. Land supply is impacted by the urban economy, such as GDP [12]. The district heads can lease a certain piece of land at a certain time according to the planning and development of the district.

The location attributes consider the location of the leased residential land, and the distances of the land to city center, district center, subway station, highway, university, high school, park and industrial park. The transportation status and the location of the leased land are crucial for the land leasing behavior of the government. Tiananmen Square is regarded as the center of Beijing. The location of the government of the district is regarded as the district center. The subway and highway are basic transport infrastructure, and can provide excellent conditions for residential development. University and high school show the educational level of a district, which can also have significant impacts on land price and house price [15]. For the key high schools, currently in China, school district housing becomes more and more popular among parents, thereby giving their children high-quality education in a more favorable atmosphere. Parks can supply an excellent environment and be of convenience for the nearby residents, and usually increase the house price [31,32]. The industrial parks have negative impacts on residential land price, because generally the residences are far from industrial parks due to the great noise and pollution.

These factors mentioned above consider the macro level of a district and micro level of a piece of leased land, such as land attributes, district-level economy, geographic location of the leased land and so on, and all possible factors are included based on the literature. These factors will be regarded as the variables of the mathematical models to study the factors affecting the behavior of the district government for leasing residential land.

### 3.2. Mathematical Model of Leased Residential Land Price

Based on above variables, the mathematical models for the price and the total area of leased residential land are presented. Similar to the hedonic model, the mathematical models are proposed based on the linear regression, and the OLS method is applied to obtain the coefficients of the models. The VIF is tested to avoid multicollinearity among the independent variables, so that the VIFs are smaller than 7.

The mathematical model of the price of leased residential land is

$$\ln LP_{it} = a + \sum_{m=1}^{M} \alpha_m A_{mit} + \sum_{j=1}^{J} \beta_j D_{jit} + \sum_{k=1}^{K} \gamma_k L_{kit} + Yr\_dum + Dis\_dum + u_{it} \quad (1)$$

where $LP_{it}$ denotes the price of each piece of leased residential land; $A_{mit}$, $D_{jit}$ and $L_{kit}$ denote the land attributes, the district-level attributes and the location attributes, respectively; $M$, $J$ and $K$ are the numbers of the land attributes, the district-level attributes and the location attributes, respectively; $Yr\_dum$ and $Dis\_dum$ denote dummy variables of the year and the district, respectively; $i$ is the district, $t$ is the year, $a$ is a constant, and $u_{it}$ denotes the error term. The district and the year may influence the results; thus, the district dummy and year dummy variables are added to control the variables of years and districts. In Table 1, a detailed description of the variables in Equation (1) is presented.

**Table 1.** The definitions and summary statistics of the variables for the price model.

| Category | Variable | Description | Obs. | Mean | Std. Dev. |
|---|---|---|---|---|---|
| | $\ln(LP_{it})$ | Transaction price of every piece of leased land of district $i$ in year $t$.(Yuan) | 187 | $7.95 \times 10^8$ | $8.99 \times 10^8$ |
| $A_{1it}$ | $\ln(Area_{it})$ | Built areas of the land leasing for district $i$ in year $t$. (m$^2$) | 187 | 75,839.24 | 69,095.43 |
| $A_{2it}$ | $\ln(FAR_{it})$ | FAR of the land for district $i$ in year $t$. | 187 | 2 | 0.72 |
| $A_{3it}$ | $MOD1_{it}$ | Land leasing mode for district $i$ in year $t$ =1 when the mode is listing; =0 otherwise. (Auction is for comparison.) | 187 132 55 | 0.71 | 0.46 |
| $A_{4it}$ | $MOD2_{it}$ | Land leasing mode for district $i$ in year $t$ =1 when the mode is tender; =0 otherwise. (Auction is for comparison.) | 187 54 133 | 0.29 | 0.45 |
| $A_{5it}$ | $LOC1_{it}$ | Location of district $i$ =1 when $i$ is center district; =0 otherwise. (County is for comparison.) | 187 59 128 | 0.32 | 0.47 |
| $A_{6it}$ | $LOC2_{it}$ | Location of district $i$ =1 when $i$ is suburban district; =0 otherwise. (County is for comparison.) | 187 114 73 | 0.61 | 0.49 |
| $D_{1it}$ | $\ln(GDP_{it})$ | GDP for district $i$ in year $t$. (Yuan) | 187 | $8.51 \times 10^{10}$ | $9.55 \times 10^{10}$ |
| $D_{2it}$ | $DM_{it}$ | Term of office for district head of district $i$ in year $t$. | 187 | 4 | 1.91 |
| $L_{1it}$ | $\ln(TA_{it})$ | Distance of the land for district $i$ in year $t$ to Tiananmen Square. (m) | 187 | 28,788.91 | 18,217.55 |
| $L_{2it}$ | $\ln(GOV_{it})$ | Distance of the land for district $i$ in year $t$ to the government of district $i$. (m) | 187 | 8752.82 | 9588.75 |
| $L_{3it}$ | $\ln(SUB_{it})$ | Distance of the land for district $i$ in year $t$ to the nearest subway. (m) | 187 | 13,181.63 | 15,094.78 |
| $L_{4it}$ | $\ln(HIGH_{it})$ | Distance of the land for district $i$ in year $t$ to the nearest highway. (m) | 187 | 3398.24 | 5569.69 |
| $L_{5it}$ | $\ln(UNI_{it})$ | Distance of the land for district $i$ in year $t$ to the nearest university. (m) | 187 | 14,376.65 | 15,060.32 |
| $L_{6it}$ | $\ln(HS_{it})$ | Distance of the land for district $i$ in year $t$ to the nearest key high school. (m) | 187 | 6099.26 | 6151.21 |
| $L_{7it}$ | $\ln(PAR_{it})$ | Distance of the land for district $i$ in year $t$ to the nearest park. (m) | 187 | 5681.05 | 3802.66 |
| $L_{8it}$ | $\ln(IP_{it})$ | Distance of the land for district $i$ in year $t$ to the nearest industrial park. (m) | 187 | 3603.74 | 4155.92 |

The vector form of Equation (1) is

$$\ln LP_{it} = a + \boldsymbol{a}\boldsymbol{A} + u_{it} \tag{2}$$

where

$$\boldsymbol{a} = (\alpha_1, \alpha_2, \ldots, \alpha_M, \beta_1, \beta_2, \ldots, \beta_J, \gamma_1, \gamma_2, \ldots, \gamma_K, 1, 1) \tag{3}$$

is the coefficient vector, and

$$\boldsymbol{A} = (A_{1it}, A_{2it}, \ldots, A_{Mit}, D_{1it}, D_{2it}, \ldots, D_{Jit}, L_{1it}, L_{2it}, \ldots, L_{Kit}, Yr\_dum, Dis\_dum)^{\mathrm{T}} \tag{4}$$

is the variable (or factor) vector.

Then we know that the expectation and variance equal to zero, i.e.,

$$\mathrm{E}(\ln LP_{it} - a - \boldsymbol{a}\boldsymbol{A}) = 0 \tag{5}$$

$$\mathrm{E}(A_j(\ln LP_{it} - a - \boldsymbol{a}\boldsymbol{A})) = 0 \tag{6}$$

where $A_j$ is the weight coefficient.

Considering the sampling data, we define the estimator $\hat{\boldsymbol{a}}$ of the vector $\boldsymbol{a}$ as

$$\hat{\boldsymbol{a}} = (\hat{a}_1, \hat{a}_2, \ldots, \hat{a}_M) \tag{7}$$

From Equations (5) and (6) we have

$$\frac{1}{N}\sum_{n=1}^{N}(\ln LP_{itn} - \hat{a} - \hat{\boldsymbol{a}}\boldsymbol{A}_n) = 0 \tag{8}$$

$$\frac{1}{N}\sum_{n=1}^{N} A_{jn}(\ln LP_{itn} - \hat{a} - \hat{\boldsymbol{a}}\boldsymbol{A}_n) = 0 \tag{9}$$

where $N$ denotes the number of sample data.

From Equation (8) we obtain

$$\bar{y} = \hat{a} + \hat{\boldsymbol{a}}\bar{\boldsymbol{A}}_n \tag{10}$$

then

$$\hat{a} = \bar{y} - \hat{\boldsymbol{a}}\bar{\boldsymbol{A}}_n \tag{11}$$

where

$$\bar{y} = \frac{1}{N}\sum_{n=1}^{N}\ln LP_{itn} \tag{12}$$

$$\bar{\boldsymbol{A}}_n = \frac{1}{N}\sum_{n=1}^{N} A_n \tag{13}$$

In Equation (9), considering the arbitrariness of sample number, we can obtain

$$\sum_{n=1}^{N} A_{jn}(\ln LP_{itn} - \hat{a} - \hat{\boldsymbol{a}}\boldsymbol{A}_n) = 0 \tag{14}$$

From Equations (11) and (14) we have

$$\sum_{n=1}^{N} A_{jn}(\ln LP_{itn} - (\bar{y} - \hat{\boldsymbol{a}}\bar{\boldsymbol{A}}_n) - \hat{\boldsymbol{a}}\boldsymbol{A}_n) = 0 \tag{15}$$

i.e.,

$$\sum_{n=1}^{N} A_{jn}(\ln LP_{itn} - \bar{y}) = \hat{\boldsymbol{a}}\sum_{n=1}^{N} A_{jn}(\boldsymbol{A}_n - \bar{\boldsymbol{A}}_n) \tag{16}$$

By solving Equation (16) we have

$$\hat{a}_i = \frac{\sum\limits_{n=1}^{N} A_{jn}(\ln LP_{itn} - \overline{y})}{\sum\limits_{n=1}^{N} A_{jn}(A_n - \overline{A}_n)} \tag{17}$$

Then we can obtain the estimator $\hat{a}$.

### 3.3. Mathematical Model of Leased Residential Land Area

The mathematical model of the total area of leased residential land is

$$\ln(LA_{it}) = a + \sum_{l=1}^{L} \alpha_l M_{lit} + Yr\_dum + Dis\_dum + u_{it} \tag{18}$$

where $LA_{it}$ denotes the total area of leased residential land; $M_{lit}$ denotes the location attributes; and $L$ denotes the number of location attributes. The detailed description and statistics of the variables in Equation (18) are shown in Table 2. The corresponding estimator $\hat{a}$ in this model can be obtained in a similar way mentioned above.

**Table 2.** The definitions and summary statistics of the variables for the total area model.

| Category | Variable | Description | Obs. | Mean | Std. Dev. |
|---|---|---|---|---|---|
| | $\ln(LA_{it})$ | Built area of total leased land of district $i$ in year $t$. (m$^2$) | 216 | 65,658.12 | 130,448.1 |
| $M_{1it}$ | $\ln(DCC_{it})$ | Distance between district $i$ in year $t$ and Tiananmen Square. (m) | 216 | 17,538.98 | 17,704.8 |
| $M_{2it}$ | $Subway_{it}$ | Numbers of subway stations in district $i$ in year $t$. | 216 | 8.7 | 13.26 |
| $M_{3it}$ | $Highway_{it}$ | Numbers of highway entrances and exits in district $i$ in year $t$. | 216 | 84.3 | 103.22 |
| $M_{4it}$ | $\ln(DIP_{it})$ | Distance between district $i$ in year $t$ and the nearest industrial park. (m) | 216 | 181.21 | 744.76 |
| $M_{5it}$ | $LOC1_{it}$ | Location of district $i$ =1 when $i$ is center district; is center district; =0 otherwise. (County is for comparison.) | 216 96 120 | 0.44 | 0.50 |
| $M_{6it}$ | $LOC2_{it}$ | Location of district $i$ =1 when $i$ is suburban district; =0 otherwise. (County is for comparison.) | 216 96 120 | 0.44 | 0.50 |

The mathematical models proposed in this study for the price and the total area of leased residential land are based on multivariate regression analysis. The relationship between the price of every piece of leased residential land (or the area of total leased residential land) with each influencing factor is linear. The OLS method is used to obtain the regression formulae of the mathematical models. From mathematical theory, the OLS method is one of the most accurate mathematical methods to fit the most optimal value; thus, the correct regression results can be obtained from the formulae in this paper. Then, the validation for the mathematical models is shown theoretically.

### 4. Data and Regression Results

To obtain the regression results, the data of all districts in Beijing from 2004 to 2015 are searched or computed by the author one by one.

The basic data, such as GDP and term of office of the district head, of all the districts in Beijing, were obtained from the yearbooks of all 18 districts.

The data of leased residential land, such as transaction price, built area, FAR and leasing modes, in Beijing, were obtained from the official website of the Beijing Planning and Land Resources Bureau. The distribution of residential land leased in Beijing from 2004 to 2015 is given in Figure 2.

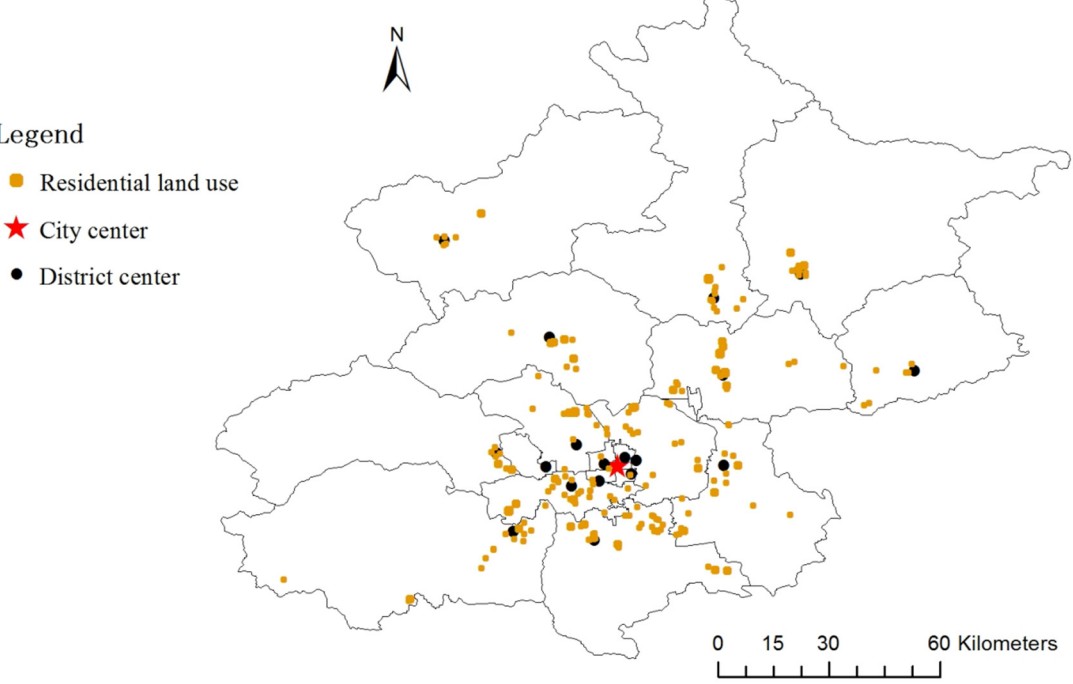

**Figure 2.** Residential land leasing from 2004 to 2015.

The longitude and latitude of all locations, such as the piece of land, the centers of Beijing and the 18 districts, subway stations, highway entrances and exits, university, high school, park and industrial park, were obtained from Google Maps. In total, 44 key high schools, 60 main universities, 50 parks and 72 industrial parks in Beijing were considered. The distributions of the subway stations and highways in 2004 and 2015 are shown in Figures 3 and 4, respectively, and the distribution of the municipal key high schools, main universities, parks and industrial parks considered in this paper is presented in Figure 5.

The distances between the location of a leased land and Tiananmen Square, district government, the nearest subway, the nearest highway entrance and exit, the nearest key high school, the nearest university, the nearest park and the nearest industrial park were computed with ArcGIS, which is geographic information system software.

After collecting the above data, from the mathematical models, the regression results of the leased residential land price and total leased area at the district level were obtained. The corresponding numerical results are presented in Tables 3 and 4.

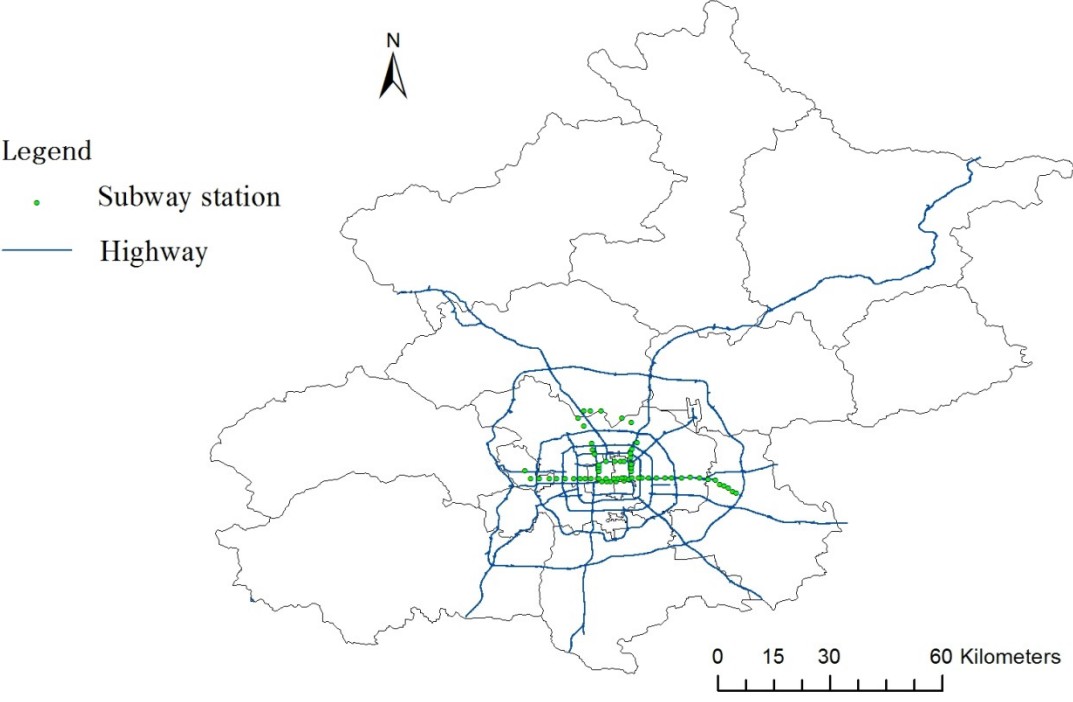

**Figure 3.** The subway stations and highways in 2004.

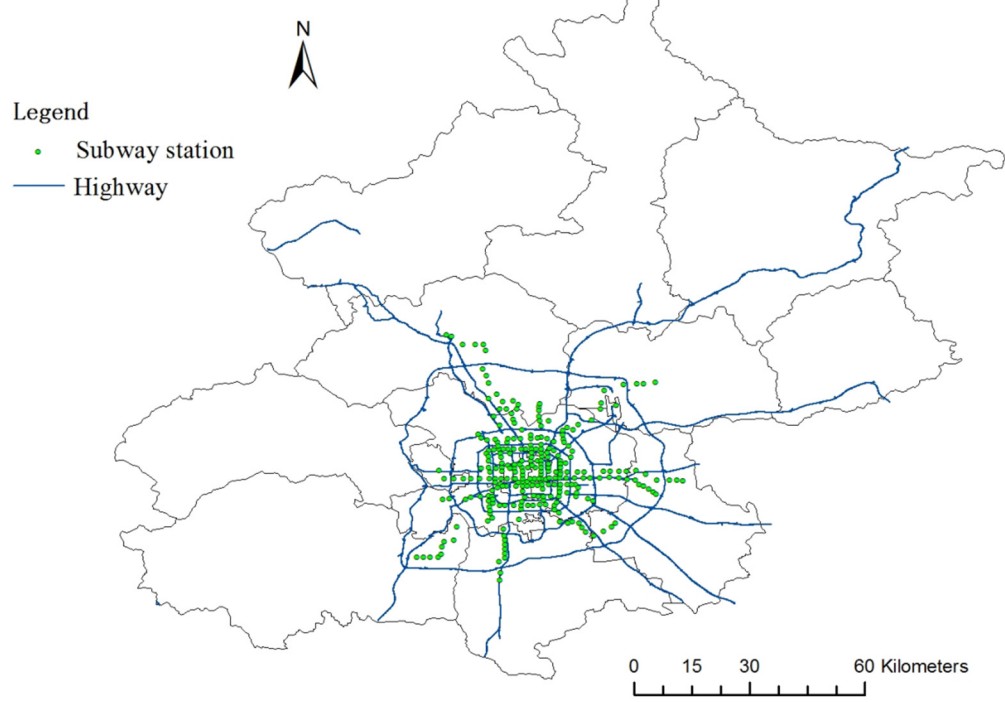

**Figure 4.** The subway stations and highways in 2015 [3].

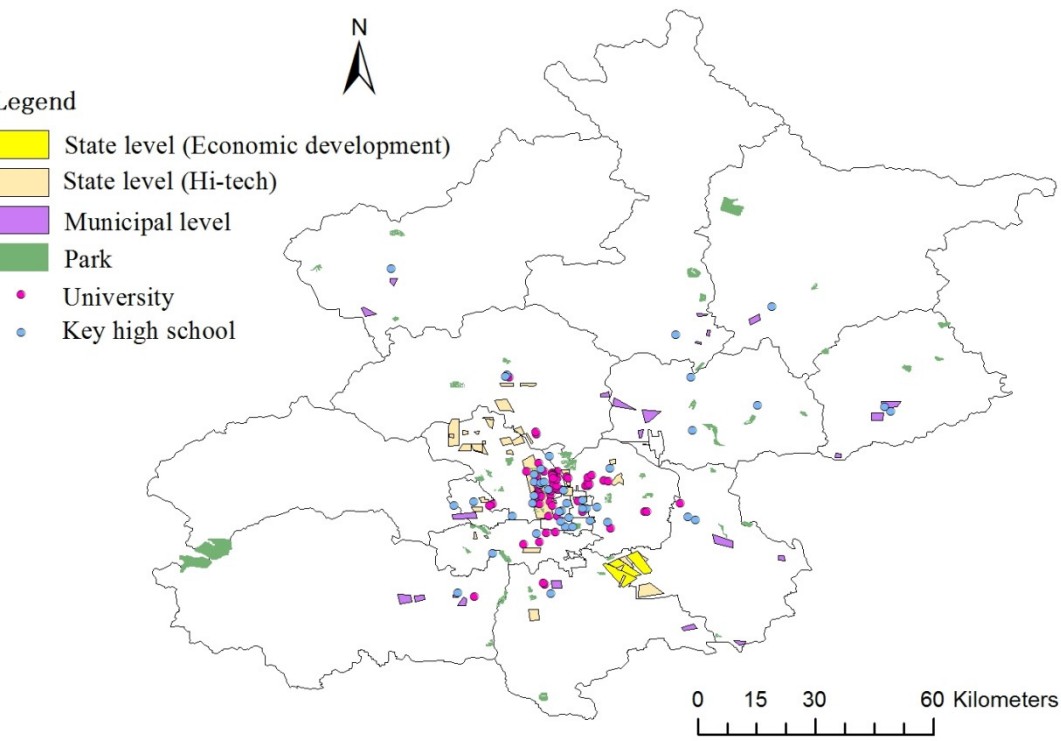

**Figure 5.** The distribution of the main universities, key high schools, parks and industrial parks.

**Table 3.** The results for the price model.

| Variable | Coefficient | Robust Standard Errors |
|---|---|---|
| Residential land attributes | | |
| $\ln(Area_{it})$ | 1.05599 *** | 0.03843 |
| $\ln(FAR_{it})$ | 0.46383 *** | 0.14348 |
| $MOD1_{it}$ | −0.16323 | 0.45901 |
| $MOD2_{it}$ | −0.06821 | 0.35298 |
| $LOC1_{it}$ | 0.05951 | 0.74614 |
| $LOC2_{it}$ | 0.41961 | 0.70880 |
| District-level attributes | | |
| $\ln(GDP_{it})$ | −0.09871 | 0.36908 |
| $DM_{it}$ | −0.03325 | 0.03341 |
| Location attributes | | |
| $\ln(TA_{it})$ | −1.02350 *** | 0.24363 |
| $\ln(GOV_{it})$ | −0.06071 | 0.05973 |
| $\ln(SUB_{it})$ | −0.05064 | 0.05986 |
| $\ln(HIGH_{it})$ | 0.09980 * | 0.05807 |
| $\ln(UNI_{it})$ | −0.05349 | 0.07592 |
| $\ln(HS_{it})$ | 0.04179 | 0.06062 |
| $\ln(PAR_{it})$ | 0.13204 ** | 0.06517 |
| $\ln(IP_{it})$ | 0.00667 | 0.01617 |
| Constant | 21.25886 ** | 8.96312 |
| District dummy | Yes | |
| Year dummy | Yes | |
| Observations | 187 | |
| $R^2$ | 0.9298 | |

Notes: ***: Significant at the 1 percent level; **: significant at the 5 percent level; *: significant at the 10 percent level.

**Table 4.** The results for the total area model.

| Variable | Coefficient | Robust Standard Errors |
|---|---|---|
| | Location attributes | |
| $\ln(DCC_{it})$ | 0.51405 ** | 0.20171 |
| $Subway_{it}$ | 0.03815 | 0.06362 |
| $Highway_{it}$ | 0.03179 | 0.05952 |
| $\ln(DIP_{it})$ | −0.20298 | 0.16756 |
| $LOC1_{it}$ | 4.36266 * | 2.58414 |
| $LOC2_{it}$ | 4.17678 *** | 1.59481 |
| Constant | −4.84172 * | 2.65913 |
| District dummy | Yes | |
| Year dummy | Yes | |
| Observations | 216 | |
| $R^2$ | 0.4677 | |

Notes: ***: Significant at the 1 percent level; **: significant at the 5 percent level; *: significant at the 10 percent level.

## 5. Discussion

Firstly, we discuss the numerical results in Table 3 for the price of residential land leasing.

For the land attributes, the area and FAR of the leased residential land are very significant. When the area of it increases by 1%, the price of the leased residential land increases by 1.06%. Then, the district government hopes to lease residential land of which the area is large, because more profits can be earned. When the FAR of the leased residential land increases by 1%, the price of the land increases by 0.46%. From an economic viewpoint, a high floor ratio indicates a high intensive land use, and then the real estate developers can make large profits from the land. Thus, the district government prefers to lease the residential land of which the FAR is high.

About the location attributes, the distance of the location of the leased land to Tiananmen Square is very significant. When the distance increases by 1%, the price of the leased residential land decreases by 1.04%. The residential land that is closer to the Tiananmen Square has a higher price, because the accessibility to the public transportation system around the city center is better, as well as the convenience level of the neighborhood. Thus, the district government tends to lease residential land near the city center.

The distances from the land to the nearest park and the nearest highway entrance and exit are significant. When the distance from the leased land to the nearest park increases by 1%, the price of the land increases by 0.13%. When the distance from the land to the nearest highway entrance and exit increases by 1%, the land price increases by 0.1%. A green space provides a healthy environment and is convenient to relax in for residents. The parks can reduce the distance between residents and nature, and are helpful for the residents to relieve stress and breathe fresh air. The highways as for transportation, and can provide the convenience for residents to travel inside and outside cities. Thus, the district government is likely to lease residential land in proximity to parks and highways.

Then, we discuss the numerical results in Table 4 for the total area of residential land leasing.

The location of the district and the distance from the district to Tiananmen Square is significant. The total area of leased residential land is larger in the center and suburban districts than that in the counties. The total area of leased residential land increases by 0.51% when the distance from the district to Tiananmen Square increases by 1%. The government leases more residential land near the city center. Residential land is also more popular in center districts and suburban districts than that in counties. The reasons may be that the high-quality public transportation system is more favorable for the residential development, as well as the convenience level of the neighborhood. The government can earn more revenues from leasing residential land in the center and suburban areas, and near the city center.

In conclusion, the land area, the FAR, the distances between the land to the city center, the nearest park and the nearest highway entrance and exit have influences on the price of



the residential land. The district location and the distance between the district and the city center affect the total area of residential land leasing in Beijing. The high land leasing price also results in great profits for the district government; thus, the district government hopes to lease the residential land with a high price. Under the control of the central government regarding house prices, the land leasing price should not increase rapidly.

According to the analysis above, we can see that the district government in Beijing hopes to lease residential land that has a large area or high FAR. Besides the land attributes, the location factors also have influences on the district government for leasing residential land. The district government prefers to lease residential land near the center of the city, park and highway; indeed, the government leases more residential land near the city center and in the center and suburban districts.

Compared with other similar studies, the residential land leasing price has a positive relationship with the area and FAR, which is in agreement with the results of Yang et al. [13]. The residential land leasing prices near the city center [13] and parks [20] are much higher, which is identical with the results in this study. It is shown that the numerical results in this paper are in agreement with those in other similar studies. The validation for the mathematical models presented in this paper is shown numerically.

Then the policy implications can be obtained from the above discussions.

Firstly, when the district government plans to lease residential land, the centers of the city and the districts should be considered as important factors to promote the pattern of combination of a center business district and high-standard residences near the center and sub-centers of the city for the development of the city in the long run.

Secondly, the district government is concerned more about the original attributes of the land, such as the land area and the FAR, when leasing residential land. Thus, the quality of the land plays an important role, and a large area and high FAR could increase profits for the governments.

Thirdly, the transportation infrastructure, such as highways, is vital for the district government to lease residential land, which is helpful for the government to develop the residences in suburban districts, promote the construction of development of suburban districts and achieve a relatively balanced residential development between the center and suburban districts.

Other developing countries implementing public ownership of land have a similar background of economic and residential development to that in China, and possibly face similar problems in residential land leasing given this background. Thus, the policy implications proposed from this study can be applied to other developing countries and other metropolitans in China, and provide a reference for them to figure out how to balance land leasing and have more sustainable development of the land use and economy.

## 6. Conclusions

Mathematical models of the price and the total area of the leased residential land are presented in this study to analyze the leasing behavior of residential land in Beijing. The factors influencing the district government's leasing behavior for residential land are considered as the variables of the models. By introducing the data of the districts in Beijing from 2004 to 2015 into the models, the numerical results of the coefficients of the influencing factors in the mathematical models are obtained by solving the equations of the regression formulae. By discussing the numerical results of the influencing factors, the district government's behavior regarding the leasing of residential land in Beijing, China, is investigated.

According to the results, area, FAR, the distance of the land to the city center, the nearest park and the nearest entrance and exit of highway affect the price of residential land; and the district location and the distance of the location of land to the city center affect total area of residential land.

According to the results in this study, the district governments in metropolitans can draw on these factors when residential land is being leased. The district governments

should also consider additional factors previously overlooked, and the district governments can formulate land policy more efficiently for sustainable development of the districts, even the whole city.

Further, some recommendations on land use for an efficient and equitable land-use policy and long-term, financially sustainable urban development are proposed. The district governments can make better land leasing decisions in the coming decades based on the results in this study. Moreover, the policy implications of this study give a reference for other large cities in China and other counties under public land ownership.

**Funding:** This research received no external funding.

**Institutional Review Board Statement:** Not applicable.

**Informed Consent Statement:** Not applicable.

**Data Availability Statement:** All data, models, or code generated or used during the study are available from the corresponding author by request: GDP and tenure of the district mayor of 18 districts in Beijing from 2004 to 2015; land leasing data for residential land use in Beijing from 2004 to 2015; data of the distances in this study; data of city center, district centers, subway stations, highways, main universities, key high schools, public parks, and industrial parks; and other data supporting this study and related statistical methodology models.

**Conflicts of Interest:** The author declares no conflict of interest.

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
