# Peer review of "Mathematical Models and Data Analysis of Residential Land Leasing Behavior of District Governments of Beijing in China"

_mathematics, doi:10.3390/math9182314_

Round 1
Reviewer 1 Report
Comments on: Mathematical models and data analysis of residential land leasing behavior of district governments of Beijing in China
This work presented the mathematical models of the price and the total area of the leased residential land. In manuscript, the variables of the mathematical models are proposed by analyzing the factors influencing the district government’s leasing behavior for residential land based on the leasing right for residential land in Beijing of China. in the mathematic work, the regression formulae of the mathematical models are obtained with the ordinary least squares method. This work is an application of mathematics in the residential land leasing behavior. The topic of the paper is interesting and the work technically sound. The reviewer thinks that the manuscript fits and is within the scope of the journal of Mathematics Journal, and the work may be of interest to its readers. Thus, the reviewer recommends a minor revision before it can be accepted.
Detailed comments are as follows.
- The developments of the ordinary least squares method, for examples, the generalized least squares method and the moving least squares method, should be discussed in Section Introduction.
- The mathematical models of the price and the total area of the leased residential land are presented in this paper. And the validation for the models needs to be shown.
- In the Tables, the font size of variable looks no consistent, please check them.
- In reference section, the format of reference should be carefully checked, for example, sometime use semicolon,and sometime use comma between authors.
- Please recheck the scale in Fig1 left map.
Reviewer 2 Report
The leasing of residential land is related to people’s life and economic development of a country. In this paper, by analyzing the factors influencing the district government’s leasing behavior for residential land in Beijing of China, the mathematical models of the price and the total area of the leased residential land are presented. Based on the data of the districts in Beijing from 2004 to 2015, the numerical results of the coefficients in the mathematical models are obtained. The numerical results show the factors the government concerned and how the factors influence the leased price and the total leased area of residential land for the large city in China. And the policy implications of district government behavior for leasing residential land in Beijing are proposed finally.
The paper can accepted for publication after considering the following points:
- Why are location dummy (LOCi), District dummy and Year dummy included in the estimation models?
- Descriptive statistics of dummy variables (such as MOD, LOC, SE) should be presented by their classifications.
Reviewer 3 Report
In this study, the mathematical models of the price and the total area of the leased residential land in Beijing are presented. By analyzing the factors influencing the district government’s leasing behavior for residential land based on the leasing right for residential land in Beijing of China, the variables of the mathematical models are proposed. The ordinary least squares (OLS) method is used to obtain the regression formulae. By imposing the data of the districts in Beijing from 2004 to 2015 into the mathematical models, the numerical results of the mathematical models are obtained. The numerical results are discussed, and the district government behavior for leasing residential land in Beijing of China is analyzed.
I recommend this paper to be accepted for publication. Some suggestions are given as follows.
- The OLS method is used in this paper. The author should discuss the development of this method in Section ‘Introduction’.
- Although analysis results are generated by the mathematical models, the reliability and casual analysis of the results need to be discussed.
